# USER-SERP INTERACTION PREDICTION THROUGH DEEP MULTI-TASK LEARNING

**Wei Jiang, Damien Jose & Gargi Ghosh**
Microsoft Bing, AI and Research
{jiwei,dajose,gghosh}@microsoft.com

## ABSTRACT

User behavior signals such as clicks are strong indicators of a search engines performance. Many existing search algorithms focus on predicting users interactions, by optimizing a relevance cost function for the query and individual web documents. The result set (list) is then generated by ranking web documents with this score. However, the probability of user interaction with a web document on a Search Engine Result Page (SERP) depends not only on a web document in isolation, but also other documents/elements present on the SERP. Our approach better predicts user interactions on web documents by not only considering the relevance of individual documents for a query, but also their interdependencies by modeling the interactions of a User on a SERP with a Multi-task Bidirectional Recurrent Neural Network (RNN).

## 1  INTRODUCTION

Optimizing an entire list of documents on the SERP can help surface diverse results and increase user satisfaction for ambiguous queries and broad intents. Search algorithms generate the document list on the SERP by rating and ranking individual documents for a given query. Understanding user behavior in web search is a well-studied area (Granka et al., 2004; Craswell et al., 2008; Dupret & Piwowarski, 2008; Guo et al., 2009; Cao et al., 2007; Xu et al., 2010; Xiong et al., 2012; Zhai et al., 2016; Borisov et al., 2016). Our work aims at predicting user interaction in the organic search domain through multi-task learning on the SERP using a Bidirectional Recurrent Neural Network and leveraging exploration for bias-reduced predictions. We use tri-letter encoding and Gradient Boosted Decision Trees to generate the feature set, inspired by the hybrid model structure presented by Facebook (He et al., 2014) and the deep structured semantic model for click prediction (Huang et al., 2013). We report Area Under Curve (AUC) and click logs based NDCG for model evaluation.

## 2  MODEL STRUCTURE

SERP query-document pairs are featurized using a feature processor and subsequently fed into a bidirectional RNN. Using a sequence of hidden embeddings from the featurization process, the bidirectional RNN generates a sequence of representations for each query-document pair. We then use a set of Multi-Layer Perceptrons (MLP), one per document, to generate the per-document click probabilities and another MLP to generate the SERP level click probability Figure 1a.

### 2.1  TEXT FEATURE ENCODING WITH DSSM

Query text and document title are encoded using a tri-letter dictionary with 49k terms, which is generated from a collection of queries and document titles on a large-scale search engine. The 49k dimension sparse term vector for both query and document title are hashed into a lower dimension of size 3k Figure 1b and then to a dense semantic feature vector of dimension 256. The relevance given a query-document pair can be measured by computing the distance between their semantic features.

## 2.2 SUPERVISED FEATURE TRANSFORMATION WITH GBDT

In addition to text features, the model injects raw custom features (e.g. number of perfect matches between query and document title) and historical user feedback features Figure 1b. As described in section 3.2, these raw custom features are encoded into sparse vectors by a set of GBDT encoders, which are pre-trained with query document pairs and clicks as labels. During feature transformation, the GBDT encoders are executed with each leaf index encoded by a one-hot encoder, per tree. The encoded 4.5k-dimensional vector is then compressed with a neural network layer with sigmoid activation, to a dense 256-dimensional vector.

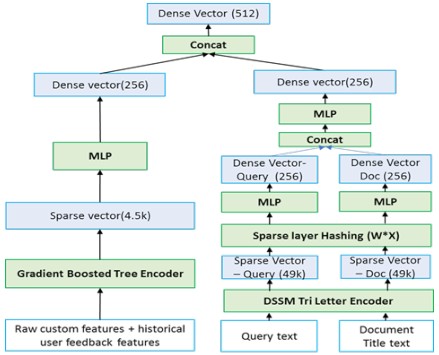

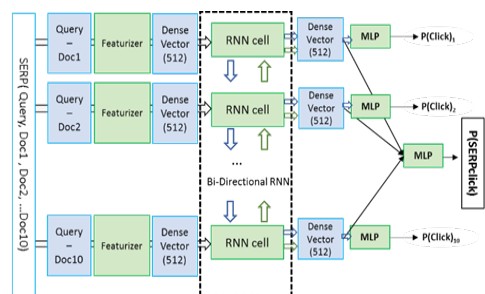

(a) Overview of the multitask deep model with featurizer and Bidirectional RNN. Each direction produces a 256-dimensional vector. Vectors from both directions are concatenated to produce a dense 512-dimensional vector, on which final MLP layers are trained.

(b) Featurizer includes GBDT feature transformation (left portion of image) and semantic feature transformation (right portion)

## 2.3 MULTITASK BIDIRECTIONAL RNN

In our proposed model (Fig 1), we use a bidirectional RNN with 3 hidden stacks to model the documents sequence in a SERP. The purpose of bidirectional is to capture the dependencies between documents, top-to-bottom and vice-versa. In this model, the input to the Gated Recurrent Unit (GRU) at each step of RNN is the concatenated representation of both numeric features and text features. The output of the stacked RNN layer is a sequence of hidden representation at each position. At each step, the output of RNN is projected to document level probabilities through MLP layers. Simultaneously, the output vector from each direction of the RNN layer goes through additional MLP layers to generate SERP level click probability. This achieves multitask learning by co-training both document level and SERP level predictions. The modeling can be described as follows:

1. Query issued by a user on the search engine is transformed to a Query vector $v_q$ by applying GBDT featurizer on query level features and DSSM featurizer on query text
2. Various SERP candidates that match the intent of the query are generated. For each document in each list a vector $(v_d)$ is generated by applying GBDT featurizer on the query-document features and DSSM featurizer on the documents title and URL text
3. The query vector and each document vector are jointly passed through MLP to produce a combined representation of query and document.
4. We apply Bi-directional RNN on the sequence of vectors from step 3, each direction of which generates a set of smaller vectors $(v_r)$
5. MLP is then applied on $v_r$ to predict the probability of the users click on the SERP

## 3 EXPERIMENTAL SETUP

### 3.1 DATA

The training (80%) and test (20%) data comprises of features and click labels extracted from uniformly sampled (0.4% daily churn) users interactions on a commerical search engine, aggregated

over one month. To generalize better over unseen data it is important use some randomized data, however the trade-off between acting optimally (exploit) vs gathering randomized data (explore) should be carefully considered to avoid regression in user experience. We use bootstrapping with Thompson's sampling for resolving the explore-exploit dilemma (Chapelle & Li, 2011).

## 3.2 FEATURES

The features, can be broadly categorized into text-based features, raw custom features from query and document information like in the MSLR dataset (Qin & Liu, 2013), and features from historical user interactions such as clicks and CTR . Raw custom features include both static features for the query, document, user etc. and dynamic features that can change over time e.g. document body, user preferences, and compound features like Okapi BM25f (Robertson et al., 2009). We mine daily document impressions and clicks and compute graded user feedback distributions conditionally (i.e.per Query*Document, Query*Domain etc.) and marginally (i.e. per Query, Document, Domain etc.) grouped by market and position on SERP. For text-based features, we use overlapping tri-letter encoding, viewed as a projection of the raw text into a semantic word embedding space. For raw numeric features we leverage tree-based transformation using GBDTs to reduce the effect of outliers and improve feature selection.

## 3.3 METRICS

**Area under the curve (AUC):** We report ROC-AUC (Area under ROC curve) at each list position as well as overall SERP. **Log loss** $\mathbb{L}$: Our model optimizes the log loss of SERP click probability, with graded relevance labels for **NDCG** computed based on the logs of click, non-click and quality of click. We normalize our scores with the per-position background Click-through-rate (CTR). We use paired t-tests for statistical significance testing with $95\%$ confidence interval and p-value of $0.05$.

## 4 RESULTS AND FUTURE WORK

Table 2a presents AUC, Log loss and NDCG measurements for our models. The baseline is a Deep Neural Network (DNN) trained to predict probability of a click for a query-document in isolation. The hidden representations were then used to predict probability of click on SERP. We present results from two models A) a DNN with GBDT and DSSM trigram encoding and B) a bidirectional RNN with GBDT and DSSM encoding, both of which are initialized by pre-training with document level click labels. Models A and B are then co-trained with SERP-level click labels achieving optimization of both document and SERP clicks. Table 2b presents a comparison of prediction accuracy/ AUC of a single item in a list, for a given position within the list. Bidirectional RNN with multitask learning achieves the best result in predicting user behavior on the SERP. Furthermore, this model is also better at predicting click probability at lower positions on the list. This is due to RNN's ability to learn the underlying sequential nature of a SERP. Model B is better than both baseline and Model A in terms of relevance on NDCG metric with very low p-values (in the order of 1e-9).

(a) SERP level AUC, Log Loss and NDCG@10

| Experiment | Model structure | AUC | Log Loss | NDCG@10 |
|---|---|---|---|---|
| Baseline | DNN | 0.773 | 0.1556 | 0.702 |
| A | DNN with GBDT and DSSM | 0.791 | 0.1491 | 0.720 |
| B | Bidirectional RNN with GBDT and DSSM | 0.813 | 0.1464 | 0.723 |

(b) Single document level AUC

| Model | Pos1 | Pos2 | Pos3 | Pos4 | Pos5 |
|---|---|---|---|---|---|
| A | 0.811 | 0.703 | 0.719 | 0.739 | 0.741 |
| B | 0.816 | 0.720 | 0.733 | 0.745 | 0.752 |

In this paper, we present a multitask RNN based framework for SERP level user behavior prediction that improves performance for both list of documents as well as individual ones. The model architecture enadles better tolerance to data and feature noise due to supervised GBDT feature transformation and text encoding. We intend to experiment with soon to be public datasets that have Query, Title and URL raw text information and more text information such as document snippet or document content. This work can be extended to model temporal sequences given time spent by users on SERPs and a search session, to improve the overall utility of user session.

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

## ACKNOWLEDGEMENTS

We would like to thank Armen Aghajanyan and Nick Crasswell {araghaja,nickcr@microsoft.com} for their help with reviewing, editing and providing valuable feedback for this paper.

