# OpenReview forum: "User-SERP Interaction Prediction through Deep Multi-task Learning"
_ICLR.cc/2018/Workshop — Reject_

### Official Review · AnonReviewer3 · 2018-03-05
**interesting use of existing techniques for user-SERP prediction**

**Rating:** 6
**Confidence:** 4

**Review:**

This paper leverages gradient boosted decision trees and deep structured semantic model from the literature  to consider inter-dependencies of documents and user interactions on a search engine result page by proposing  a multi-task bi-RNN model. Experimental results show improved scores over some baselines.

In general, the paper is well-written and the ideas are clear, but it would have been helpful to see more analyses with examples about why/how the proposed model is performing better than the baselines. Furthermore, it would be nice to see the impact of various raw custom features, trigram encoding etc. on the overall performance of the model. Also, it is not clear if the dataset is publicly available, hence, it is not possible to effectively judge the proposed method's value in comparison to other related works from the literature.

---

### Official Review · AnonReviewer2 · 2018-03-10
**Interesting application of RNNs to SERP click prediction, but quite hard to read**

**Rating:** 5
**Confidence:** 2

**Review:**

This paper presents a bidirectional RNN to predicts user clickthrough in search results.  Experiments show that the RNN approach, which considers the full SERP jointly, outperforms an approach that considers each result independently.

The paper is interesting and I think people at the conference would enjoy learning about this application of RNNs.  However, the paper is very densely packed, and the presentation of the methods is some combination of very hard to understand and erroneous.  Figure 1a says it is depicting an RNN, but it isn't (there is no RNN in that figure).  So I gather that Figure 1a is the "featurizer" shown on the lhs of Figure 1b.  This took me awhile to understand (and I may still be mistaken, because of course the figure captions say a completely different thing and never make the relationship between Figure 1a and 1b explicit).  I had a hard time following the description of the model, had to read it multiple times.  These issues with clarity make me lean toward the reject side unfortunately.

Minor:
I was unsure what the DNN baseline consisted of...does it use the same featurization as the RNN, just without the RNN?
I found the explore/exploit discussion confusing, is this clickthrough model also being used to select and rank search results for the engine?
References to figures in the text need parens around them, or some other kind of context.  Also those references seem to come at the wrong place in the text
a search engines performance -> a search engine's performance

---

### Official Review · AnonReviewer1 · 2018-03-10
**Weak accept**

**Rating:** 6
**Confidence:** 3

**Review:**

Clarity:
The paper is quite densely written. The references help clear up exactly how the GBDT and DSSM representations are made, which is the hardest part to read in the paper.
The figure text for Fig 1 is hard to read, because in (a) you describe both the left and right network, while in caption (b) you refer back to the network in (a). It would be clearer if figure text (a) and (b) described what is actually seen above them, such that (a) focuses on the featurizer, and (b) on the bidirectional RNN and prediction on click and SERP click.
In the sentence: "Table 2b presents a comparison of prediction accuracy/ AUC of a single item in a list, for a given position within the list." do you mean only AUC and not accuracy?
quality:
Quality:
The experiments are sensible, and the chosen methods seem suitable for the problem. The experimental results show the bidirectional network helps increase performance on both SERP level and document level. However, it is not verified whether the multi task objective improves performance on the separate task.

Originality:
The paper uses a standard featurizer for (query,doc) featurizing, but I have not seen Click and SERP click trained simultaneously. I am unaware if the approach of considering the whole SERP, when computing click probabilities, has been done before, but it is a very good idea and seems like a good way forward.

Significance:
There is potential to improve ranking models.

To summarize:

Pros:
Appropriate model for problem.
It seems very likely that the use of multi task learning can provide an improvement in this area.
Not looking at documents in isolation, but considering the whole SERP is a good idea.

Cons:
Explanation of the figures should be redone.
No evaluation of whether the multi task learning actually improves the individual tasks.
A better overview of the size of the data set used.

---

### Decision · Program_Chairs · 2018-03-20
**ICLR 2018 Workshop Acceptance Decision**

**Decision:**

Reject

**Comment:**

Based on the reviews, this paper has not been accepted for presentation at the ICLR workshop. However, the conversation and updates can continue to appear here on OpenReview.